# Nucleoside Phosphorylases make *N*7-xanthosine

Sarah Westarp [1,2], Felix Brandt [3], Lena Neumair [1], Christina Betz [1], Amin Dagane[1], Sebastian Kemper [4], Christoph R. Jacob [3], Peter Neubauer [1], Anke Kurreck [1,2] ✉ & Felix Kaspar [5] ✉

Modern, highly evolved nucleoside-processing enzymes are known to exhibit perfect regioselectivity over the glycosylation of purine nucleobases at *N*9. We herein report an exception to this paradigm. Wild-type nucleoside phosphorylases also furnish *N*7-xanthosine, a "non-native" ribosylation regioisomer of xanthosine. This unusual nucleoside possesses several atypical physicochemical properties such as redshifted absorption spectra, a high equilibrium constant of phosphorolysis and low acidity. Ultimately, the biosynthesis of this previously unknown natural product illustrates how even highly evolved, essential enzymes from primary metabolism are imperfect catalysts.

The modern natural nucleosides are *N*1-ribosides of pyrimidine nucleobases and *N*9-ribosides of purine nucleobases. Although, for instance, *N*7-ribosyl purines were likely readily available under pre-biotic conditions[1,2], nature elected to use *N*9-ribosides. As such, early metabolism was evolved to selectively furnish, degrade and re-use *N*9-ribosyl purines. As a result, modern (highly evolved) nucleoside-processing enzymes generally exhibit perfect control over the stereoselectivity of the attack at the anomeric center as well as the regioselectivity of the glycosylation. Although unusual glycosylation isomers of purine nucleobases are known in plants (where they serve as hormones)[3,4], the other conceivable ribosylation isomers of the natural nucleobases do not occur as regular products of metabolism.

In modern nucleoside metabolism, nucleoside phosphorylases (NPs) are responsible for recycling nucleosides by cleaving them into the corresponding nucleobase and ribose 1-phosphate (**Rib1P**), which is funneled back into primary carbon metabolism. These enzymes are ubiquitous across all kingdoms of life and known for their high selectivity and robustness[5–7]. As their transformations are tightly thermodynamically controlled[8], they also catalyze the corresponding reverse reactions (glycosylations) and thereby maintain balanced physiological concentrations of nucleosides and nucleobases. Any failure to do so generally results in disease (e.g. immune disorders)[9]. Thus, modern NPs are under considerable selection pressure to execute their physiological tasks fast and selectively. As group-specific enzymes, purine NPs are faced with the additional challenge of recognizing multiple different purine nucleosides (and nucleobases) as substrates and catalyzing transformations from and towards the natively occurring *N*9-ribosyl purines. Although purine NPs operate with astonishing selectivity, their ability to promiscuously degrade "non-native" glycosylation regioisomers of purines is documented by a handful of examples[10–13]. However, imperfect regioselectivity in glycosylation reactions with naturally occurring purines has not been reported yet for purine NPs.

Here, we report an exception to this paradigm and show that NPs can furnish *N*7-xanthosine (**N7X**, Fig. 1a), a "non-native" glycosylation regioisomer of xanthosine. Since *N*7-ribosides such as **N7X** are not known as natural products and remain uncharacterized, we herein report on the discovery of this unusual nucleoside as well as its atypical spectroscopic and physicochemical properties.

## Results

During our ongoing efforts to facilitate nucleoside synthesis with NPs[14–16], we serendipitously discovered **N7X** as an unexpected byproduct of a reaction cascade involving a bacterial purine NP and a guanine deaminase (see Supplementary Fig. 1). With the former enzyme cleaving guanosine into guanine and **Rib1P** and the latter

[1]Chair of Bioprocess Engineering, Institute of Biotechnology, Faculty III Process Sciences, Technische Universität Berlin, Ackerstrasse 76, 13355 Berlin, Germany. [2]BioNukleo GmbH, Ackerstraße 76, 13355 Berlin, Germany. [3]Institute of Physical and Theoretical Chemistry, Technische Universität Braunschweig, Gaußstraße 17, 38106 Braunschweig, Germany. [4]Institute for Chemistry, Technische Universität Berlin, Straße des 17. Juni 135, 10623 Berlin, Germany. [5]Institute for Biochemistry, Biotechnology and Bioinformatics, Technische Universität Braunschweig, Spielmannstraße 7, 38106 Braunschweig, Germany. ✉e-mail: anke.wagner@tu-berlin.de; felix.kaspar@web.de

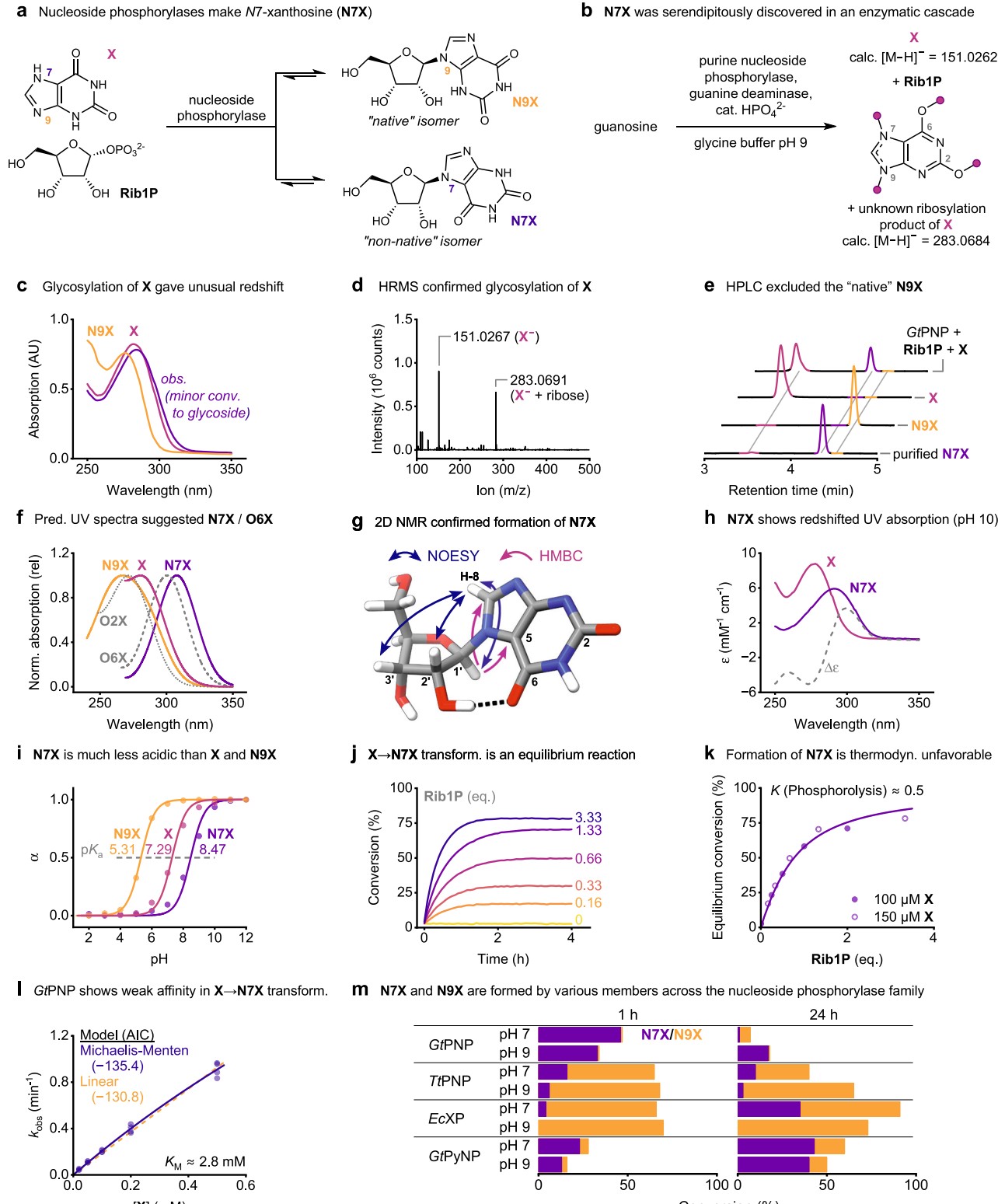

**Fig. 1 | Discovery and characterization of N7X. a** Formation of **N7X** and/or **N9X** catalyzed by nucleoside phosphorylases. **b** Serendipitous discovery of **N7X** in an enzymatic cascade through (**c**) its unprecedented redshift compared to the free nucleobase. **d**–**g** Orthogonal data confirming **N7X** as the major product of the glycosylation of **X** by *Geobacillus thermoglucosidasius* purine NP (*Gt*PNP) at pH 9. **f** Energy-minimized structure of **N7X** and key 2D NMR correlations. **h** UV absorption spectra of **X** and **N7X** at pH 10, where both species exist as monoanions. **i** p$K_a$ values as determined by UV spectroscopy. **j**, **k** Continuous monitoring of the **X** → **N7X** glycosylation at pH 10 (with 0.4 g L$^{-1}$ *Gt*PNP) to establish its equilibrium constant. By convention, these equilibrium constants are given for the phosphorolytic direction. **l** Michaelis-Menten kinetics for **N7X** formation by *Gt*PNP. The relationship between $k_{obs}$ and [**X**] is non-linear (as supported by the Akaike information criterion, AIC), allowing us to estimate a Michaelis-Menten constant. **m** Formation of **N9X** and **N7X** by various wild-type nucleoside phosphorylases (with 500 μM **X** and 1 g L$^{-1}$ enzyme). The bars in this panel each display the isomer distribution obtained in a single experiment ($n = 1$). Please see the Supplementary Information for experimental details and the externally hosted supplementary material for all raw data[38].

quantitatively hydrolyzing guanine into xanthine (**X**), the expected products of this sequence should be **Rib1P** and **X** (Fig. 1b). As *N*9-xanthosine (**N9X**, the "native" isomer) is typically not converted by bacterial nucleoside phosphorylases to any significant extent (this process requires a specialized enzyme: xanthosine phosphorylase), the reverse reaction, glycosylation of **X**, similarly should not proceed. Thus, **Rib1P** and **X** should be terminal products of this reaction sequence. However, UV absorption spectra obtained during this transformation proved inconsistent with this assumption (Fig. 1c). Similar results were obtained when **X** and **Rib1P** were directly incubated with the purine NP. We consistently observed absorption at higher wavelengths than anticipated for **X** and HPLC analysis of the reaction mixture showed a single additional peak corresponding to an unknown product (Fig. 1e). HRMS clearly supported the formation of a riboside of **X** but the retention times and absorption spectra of the reaction product evidently did not match **N9X** (Fig. 1c–e). We subsequently consulted DFT calculations to predict dominant tautomers and their UV spectra, which suggested **N7X** and *O*6-xanthosine (**O6X**) as sufficiently redshifted candidates, excluding *O*2-xanthosine (**O2X**, Fig. 1f, please see the Supplementary Information and Supplementary Fig. 3 for details). A detailed NMR-spectroscopic analysis of a partially purified sample of the reaction product in $D_2O$ revealed **N7X** as the likely product. A weak HMBC correlation between the anomeric position on the ribosyl unit and C5 of the **X** unit, as well as a strong correlation to C8, make *N*7 the most likely ribosylation site. NOESY correlations between the ribosyl hydrogens (1′, 2′ and 3′) and H-8 support this assignment (Fig. 1g). Subsequent freeze-drying of the sample, trituration with DMSO-$d_6$ and comparison to Bridson's 1998 NMR data (the only reported synthesis of **N7X**)[17] unambiguously supported the formation of **N7X** (Supplementary Table 3). Notably, **N7X** can be distinguished well from **N9X** by $^1$H NMR through its significantly greater shift for H-8 and H-1′ as well as the much smaller H-1′,H-2′-coupling constant (Supplementary Table 4), likely caused by slight ring torsion introduced by intramolecular H-bonding (Fig. 1g).

**N7X** has several unusual physicochemical properties. First, **N7X** is the only known natural nucleoside whose UV absorption spectra are redshifted compared to those of the free nucleobase (Fig. 1h)[18]. **N7X**'s sweeping absorption spectra (particularly those of the anion, Supplementary Fig. 4) are more reminiscent of free pyrimidine nucleobases than purine nucleosides. Secondly, **N7X** has a p$K_a$ of around 8.5 (Fig. 1i and Supplementary Fig. 4), as determined by UV spectroscopy, which is notably higher than free **X** (ca. 7.3) and **N9X** (ca. 5.3)[19]. Both of these properties may originate from the unique electronic situation in **N7X**'s nucleobase. Although DFT calculations suggest that the negative charge in the anions of both **N7X** and **N9X** predominantly resides at *O*2 (Supplementary Fig. 3), only **N7X** can form an intramolecular H-bond between *O*6 and the 2′-OH (Fig. 1f). This grants **N7X**'s nucleobase more aromatic character and leads to a pronounced redshift in its absorption spectra, in line with the trends exhibited by pyrimidine nucleosides and -bases[18]. Similar, although much less pronounced, trends have been described for analogues of 8-azaguanosine[20]. Thirdly, **N7X** displays an equilibrium constant of phosphorolysis of ca. 0.5 at pH 10, which we determined by continuously monitoring the glycosylation of **X** by UV spectroscopy using principles of spectral unmixing (Fig. 1j, k)[8,18,21,22]. This unusually high value could be qualitatively confirmed by reactions monitored discontinuously by HPLC (Supplementary Table 4). As purine nucleosides typically have equilibrium constants of phosphorolysis of 0.01 – 0.1 (irrespective of the sugar moiety)[8,23], this makes **N7X**'s phosphorolysis the most exothermic one known among purines. The next closest equilibrium constants of phosphorolysis of purines are those of **N9X** (0.2, determined by HPLC, Shugar reported 0.05)[19] and inosine (0.1, determined by discontinuous spectral unmixing)[8].

To our surprise, non-selective formation of **N7X** is a common reactivity across the entire family of NPs (Fig. 1m). We made our initial discovery using the purine NP from *Geobacillus thermoglucosidasius* (*Gt*PNP), which almost exclusively furnishes **N7X** under alkaline conditions. Compared to its native reactions (e.g. the phosphorolysis/glycosylation of adenosine or guanosine), which proceed with rate constants of >1 s$^{-1}$, the formation of **N7X** is a slow process. Under alkaline conditions, this transformation happens well over an order of magnitude slower and with a $K_M$ value exceeding the solubility limit of free **X**, as determined from kinetic experiments assessed by UV spectroscopy (Fig. 1l). Interestingly, at neutral pH values, this enzyme also yields **N9X** as a product of the same reaction and slowly degrades **N7X** (likely via hydrolysis, Fig. 1m). This process is pH-dependent (it does not happen at pH 10 to a relevant degree, Fig. 1j) and enzyme-catalyzed (purified **N7X** is stable, Supplementary Fig. 6). Other purine NPs, such as the ones from *Thermus thermophilus* (*Tt*PNP), *Escherichia coli* and human, also make both **N7X** and **N9X** in different ratios, as do the xanthosine phosphorylase from *E. coli* (*Ec*XP) and the pyrimidine NP from *G. thermoglucosidasius* (*Gt*PyNP, albeit much slower, Fig. 1m and Supplementary Table 4). Although the elucidation of the exact mechanisms underpinning the observed selectivity preferences are beyond the scope of this work, we hypothesize that differences in H-bonding behavior may be responsible. Purine NPs preferentially yielding **N7X** possess a nucleobase-coordinating aspartate, while those preferentially making **N9X** have an asparagine at the analogous position (Supplementary Fig. 5).

In conclusion, we serendipitously found that NPs glycosylate **X** to yield both **N7X** and **N9X** in an enzyme- and pH-dependent fashion. **N7X** is a stable nucleoside, although its formation is thermodynamically disfavored and some purine NPs also catalyze its hydrolysis. We assume that the formation of **N7X** among NPs has previously gone unnoticed because i) **N7X** and **N9X** are indistinguishable by mass spectrometry, ii) **N7X** only forms slowly under somewhat unusual conditions (alkaline media and comparably high concentrations of **Rib1P** and **X**), and iii) NPs are usually screened in the phosphorolysis direction (obscuring potential glycosylation isomers only occurring in the reverse reaction). However, the synthesis of **N7X** by wild-type NPs likely carries no physiological relevance since both free **X** and **Rib1P** only occur in low micromolar concentrations in vivo (far below the $K_M$ values of this slow transformation)[24], which precludes this transformation from happening to any relevant degree. This lack of physiological relevance is likely also responsible for the retention of this promiscuous activity in modern NPs. Collectively, this case study provides a notable exception to the paradigm that "native" purine nucleosides are *N*9-ribosides and further illustrates how even highly evolved enzymes are imperfect catalysts. Although catalytic promiscuity is an inherent trait of proteins[25-27], it is comparably rare to observe a condition-dependent preference for the formation of a "non-native" reaction product in enzymes under evolutionary pressure for high selectivity. As NPs otherwise operate with perfect selectivity in their native target transformations, we hope that this report sparks future research into other "non-native" nucleoside isomers as well as the implications of this promiscuity for early metabolism and the convergent evolution of NPs[6].

## Methods

### Accessibility statement

The data presented in this manuscript are depicted using scientific color maps. Since ca. 4% of the human population are color vision-deficient, we made a conscious effort to avoid unscientific uses of color, such as color ambiguities and other biases that could lead to misrepresentation, limited accessibility, or loss of information upon reduction of the color space[28,29]. Thus, all figures herein were created using the scientific color map plasma, which retains color-coded information for all readers.

## Protein production and purification

All enzymes were heterologously produced in *Escherichia coli* as His$_6$-tagged proteins through isopropyl β-D-1-thiogalactopyranoside (IPTG)-induced overexpression, as previously described[21]. For protein production, we employed two different strategies:

i) For production in Enpresso B medium (Enpresso GmbH, Berlin, Germany), which simulates a fed-batch culture in shake flasks, cells were washed from an agar plate of the expression strain (*E. coli* BL21g harboring the respective plasmid) with 1 mL 0.9% NaCl using a spatula, according to the manufacturer's standard protocol. Then, OD$_{600}$ was measured and 50 mL Enpresso B medium supplemented with antibiotic (100 mg L$^{-1}$ ampicillin) was inoculated to an initial OD$_{600}$ of 0.15. The resulting culture was grown at 37 °C and 250 rpm for 17 h before expression was induced by addition of IPTG to a final concentration of 0.1 mM (50 μL of a 0.1 M stock). The culture was incubated at 30 °C and 250 rpm for 24 h before the cells were harvested by centrifugation (8000 g, 10 min, 4 °C). The resulting cell pellet was either stored at −20 °C until further use or immediately subjected to lysis and purification.

ii) Alternatively, for *Gt*PNP and *Gt*PyNP, we employed a production workflow in traditional TB medium. To this end, a 20 mL preculture of the expression strain (*E. coli* JM109 harboring the respective plasmid) was grown in LB medium supplemented with antibiotic (50 mg L$^{-1}$ ampicillin) overnight at 37 °C. This preculture was used to inoculate 250 mL TB medium (12 g L$^{-1}$ tryptone, 24 g L$^{-1}$ yeast extract, 5 g L$^{-1}$ glycerol, 2.31 g L$^{-1}$ KH$_2$PO$_4$, 12.54 g L$^{-1}$ K$_2$HPO$_4$) containing antibiotic to an initial OD$_{600}$ of around 0.15. This culture was incubated at 37 °C and 200 rpm until an OD$_{600}$ of > 0.6 was reached, which typically happened after 2 – 3 h. At this point, protein production was induced by adding IPTG to a final concentration of 0.1 mM (25 μL of a 1 M stock in water). Expression was carried out overnight (ca. 20 h) at 37 °C and cells were harvested by centrifugation (4000 g, 20 min, 4 °C). The resulting cell pellet was either stored at −20 °C until further use or immediately subjected to lysis and purification.

For lysis and purification, pelleted cells were resuspended in binding buffer (20 mM KH$_2$PO$_4$, 500 mM NaCl, 20 mM imidazole) to a concentration of around 0.5 g$_{pellet}$ mL$^{-1}$ and Pierce™ protease inhibitor mini tables (Thermo Fisher Scientific) were added (one per 10 mL of suspension volume). Cells were disrupted by sonication (6 min, 10 s pulse, 10 s breaks, 60% amplitude or 10 min, 30 s pulse, 30 s breaks, 30% amplitude – both protocols performed similarly well). For some enzymes, the resulting lysate was heated for 20 – 30 min to precipitate *E. coli* proteins (60 °C for *Gt*PNP and *Gt*PyNP, 80 °C for *Tt*PNP). Cell debris and precipitated protein was then removed by centrifugation (11000 rpm, 45 min, 4 °C) and filtration (0.45 μm pore size). The cell free extract was applied to a Ni Sepharose Histrap™ column (GE Healthcare) preequilibrated with binding buffer. Non-specifically bound proteins were removed by washing with approximately 10 column volumes (CV) of binding buffer and the target protein was eluted with elution buffer (20 mM KH$_2$PO$_4$, 500 mM NaCl, 500 mM imidazole) over ca. 3 CV. Fractions containing pure target protein (as assessed by SDS PAGE) were combined and, if necessary, concentrated by centrifugation (Vivaspin, Sartorius, Göttingen, Germany, molecular weight cut-off at 10 kDa). Afterwards, the protein was desalted into 20 mM MOPS buffer (pH 7, adjusted at 20 °C) or 20 mM taurine buffer (pH 9, adjusted at 20 °C) using a PD-10 desalting column (GE Healthcare). This desalting procedure was repeated once (for a total of two passes through a PD-10 column per protein preparation). Alternatively, the target protein obtained from the affinity column was directly dialyzed against 2 mM phosphate buffer (pH ≈ 9). The resulting protein preparations were stored at 4 °C. Alternatively, glycerol was added to a final concentration of 50% (v/v) to store the resulting protein stocks at −20 °C. Typical stock concentrations of pure protein ranged from 1 to 10 g L$^{-1}$ (calculated with 1 AU at 280 nm being equal to a protein concentration of 1 g L$^{-1}$).

## Discovery of N7X

We discovered **N7X** via its unusual UV spectroscopic properties. In a reaction cascade featuring a purine nucleoside phosphorylase (purine NP, converting guanosine to guanine) and a deaminase (converting guanine to xanthine), we recorded UV spectra which did not match the expected products. For instance, we performed reactions with 200 μM guanosine, 400 μM phosphate, 100 μg mL$^{-1}$ (ca. 3.8 μM) *Gt*PNP and 1 μg mL$^{-1}$ deaminase in 50 mM glycine buffer pH 9 in a total volume of 500 μL at 40 °C for 2 h. Then, 100 μL of reaction mixture were quenched in 100 μL 200 mM NaOH directly in wells of a UV-transparent 96-well plate. Subsequently, we recorded UV absorption spectra from 250 to 350 nm in steps of 1 nm with a platereader. Reference spectra for **X** and **N9X** were obtained from solutions of either compound in 50 mM glycine buffer and analogous dilution in aq. NaOH. This gave spectra such as those shown in Fig. 1c.

To check for the potential formation of glycosides of **X**, an analogous reaction mixture to the one described above was quenched 1:1 with MeOH, centrifuged to remove any precipitate (13,000 rpm, 10 min), and subjected to HRMS analysis (negative mode, as all relevant components in this mixture are anions at pH 9). This yielded clear evidence for the formation of a glycoside (Fig. 1d) with ions corresponding to xanthosine (**N9X**) or any of its isomers.

## HPLC analyses

To further exclude the formation of **N9X** by *Gt*PNP at pH 9, we analyzed the reaction products by HPLC. To this end, we performed a reaction containing 500 μM **X**, 500 μM **Rib1P**, and 1 g L$^{-1}$ *Gt*PNP in 50 mM glycine-KOH buffer pH 9. This reaction was started by the addition of the enzyme and incubated at 50 °C for 24 h. Afterwards, a 50 μL sample of the reaction mixture was quenched by addition to 50 μL ice cold methanol. The resulting sample was centrifuged (13,000 rpm, 10 min) and analyzed by HPLC, using an Agilent 2200 system with DAD detector at 260 nm. The eluents were a) 50 mM ammonium-acetate buffer pH 5 (from a 1 M stock adjusted to the correct pH with 100% acetic acid) and b) MeCN. The column was a 250 × 4.6 5 μm Kinetex Evo C18 Core Shell (Phenomenex), used at a flow rate of 1 mL min$^{-1}$ and heated to 25 °C. The elution method consisted of a linear gradient from 3 to 40% MeCN over 10 min, followed by 4 min of re-equilibration at 3% MeCN. This experiment provided further evidence that the reaction product of the *Gt*PNP-catalyzed glycosylation of **X** at pH 9 was not **N9X**. Indeed, under these HPLC conditions, the typical retention times were 3.4 – 3.5 min (**X**), 4.3 – 4.4 min (**N7X**), and 4.5 – 4.6 min (**N9X**), as confirmed with authentic standards of **X** and **N9X**, and purified **N7X**.

## Computational prediction of UV absorption spectra

To narrow down possible glycosylation isomers of **X**, we first predicted UV spectra for all likely candidates by density functional theory (DFT). To this end, DFT calculations for all structures (Supplementary Fig. 3) were performed using the Amsterdam Modeling Suite (AMS) 2020.203[30]. For initial geometry optimizations the PBE[31] XC functional with a TZP slater-type orbital basis set[32] was applied and the COSMO[33] model was used for solvation in water. The subsequent time-dependent DFT (TDDFT) single-point calculations were achieved using the B3LYP[34–36] hybrid functional with a QZ4P basis set[32]. Solvation in water was again performed by COSMO and excitation energies were calculated according to the Davidson algorithm[37]. The generation of UV/Vis spectra was achieved in Python. For each spectrum, the contribution of the different calculated structures to the spectrum was determined based on the corresponding electronic energy applying the Boltzmann equation. The final spectra were subsequently convoluted applying peak broadening with Gaussians. The most redshifted peak of each spectrum was then normalized to its maximum and plotted as shown in Fig. 1f.

## Purification of N7X and NMR analyses

**N7X** was produced by *Gt*PNP-catalyzed glycosylation of **X** and purified by semi-preparative HPLC. To this end, 18.3 mg **X**, 119.2 mg **Rib1P**, and *Gt*PNP (0.1 g L$^{-1}$) were combined in 12 mL 50 mM HEPES buffer pH 7 (although pH 7 does decrease the stability of **N7X** in the presence of *Gt*PNP this enzyme is much more active in this transformation than at pH 9, see above). This equals final concentrations of 10.5 mM **X** and 21 mM **Rib1P** in the reaction. The reaction mixture was incubated at 60 °C for 5 h and then frozen to stop the reaction. Next, the mixture was thawed and the denatured enzyme and precipitated **X** were removed by centrifugation at 4 °C. **N7X** was then purified by semi-preparative HPLC using a 250 × 10 mm MultoKrom100-10 C18 column (Chromatographie Service GmbH, Langerwehe, Germany). The purification method used water (ice-cold) and MeOH (room temperature) as eluents, a flow rate of 4.7 mL min$^{-1}$ and an elution program starting with isocratic elution with 3% MeOH for 5 min, followed by a linear gradient to 10% MeOH over 2.5 min (7.5 min run time) and a drop back down to 3% MeOH over 0.5 min (8 min run time), which was held for 6 min (14 min run time) to re-equilibrate the column. Per run, 2 mL of sample were injected, for a total of 6 runs. **N7X** typically eluted between 3 and 5 min of run time, followed by **X** between 5.5 and 7.5 min. **N9X** was not detected in this experiment. This yielded a total of 12 mg partially purified **N7X**, which contained **Rib1P** and HEPES as major impurities (detected by NMR) due to its early elution from the HPLC column. Of this sample, 8 mg were dissolved in D$_2$O (containing 0.05 wt% TSP as internal standard) for NMR analysis. Following the tentative identification of the nucleoside product as **N7X** (see the Supplementary Information for structure elucidation), this NMR sample was diluted with water, freeze-dried and triturated with DMSO-d$_6$, which dissolved **N7X**, some of the HEPES, while leaving most of the **Rib1P** untouched. NMR analysis of the resulting sample unambiguously identified the product as **N7X** due to matching data with Bridson's report (Supplementary Table 3)[17]. All NMR data collected in this study are freely available from the externally hosted supplementary information[38].

## Reference UV spectra of X and N7X at pH 10

At pH 10, **X** and **N7X** almost completely exist as their anions (see Fig. 1i). We therefore obtained reference spectra of both compounds at pH 10 to enable continuous reaction monitoring (see below). To this end, we prepared a 100 µM solution of **X** in 50 mM glycine/proline buffer pH 10 and measured its UV absorption spectrum in a platereader using UV-transparent 96-well plates. The Lambert-Beer law directly provided the extinction coefficients of **X** at all wavelengths. Since accurate preparation of exact concentrations of **N7X** proved difficult, we used isosbestic normalization based on the known isometric point of glycosylation to correct for concentration errors. Thus, we prepared a ca. 100 µM solution of purified **N7X** and obtained its UV absorption spectrum in analogy to **X**. Next, we normalized the spectra of **X** and **N7X** to the isosbestic point of glycosylation (288 nm) and used the scaling factor between the extinction coefficient of **X** and the normalized spectrum of **X** to calculate the extinction coefficients of **N7X** based on the normalized spectrum of **X**. The result is shown Fig. 1h and all experimental and calculated data are available from the externally hosted supplementary information[38].

## Absorption spectra of X, N9X and N7X as functions of pH

To determine the p$K_a$ values of **X, N9X** and **N7X**, we obtained UV absorption spectra from pH 2 − 12 and applied isosbestic normalization for fitting. To this end, we prepared ≈100 µM solutions of each compound in a universal buffer mix containing citrate (p$K_a$ = 3.1, 4.8, and 6.4), HEPES (p$K_a$ = 3.0 and 7.5), asparagine (p$K_a$ = 2.1 and 8.8) and proline (p$K_a$ = 2.0 and 10.6) (all adjusted to the respective pH with HCl and NaOH, all with 5 mM final concentration of each buffer component) and obtained UV absorption spectra of these solutions from 250 to 350 nm in steps of 1 nm in a platereader using UV-transparent 96-well plates. This revealed isosbestic points of deprotonation of 274 nm (**X**), 259/268 nm (**N9X**), and 279 nm (**N7X**). All spectra were then normalized to the absorbance at the isosbestic point (Supplementary Fig. 4) and the distribution of the neutral and deprotonated species was calculated as a function of pH as described in the Supplementary Information, giving the fits shown in Fig. 1i.

## Equilibrium constant of phosphorolysis of N7X

To determine the equilibrium constant of phosphorolysis of **N7X**, we performed glycosylation experiments monitored by UV. To this end, we continuously monitored the glycosylation of **X** catalyzed by *Gt*PNP at pH 10, where **X** and **N7X** almost exclusively exist as monoanions and *Gt*PNP selectively yields **N7X** (and not **N9X**). Specifically, we performed glycosylation reactions with **X** (100 or 150 µM), **Rib1P** (0, 25, 50, 100, 200 or 500 µM, equivalent to 0, 0.25, 0.5, 1, 2 and 5 or 0.16, 0.33, 0.66, 1.33 and 3.33 equivalents) and 400 µg mL$^{-1}$ *Gt*PNP (equivalent to ca. 15.4 µM) in 50 mM glycine/proline buffer (containing 50 mM of both buffer components) pH 10 in wells of a UV-transparent 96-well plate in a total volume of 200 µL at room temperature (ca. 21 °C). These reactions were started by addition of 150 µL reaction mixture (containing all reagents except **Rib1P**) to 50 µL appropriately diluted solution of **Rib1P**. These reactions were monitored at 288 nm (the isosbestic point of phosphorolysis of this transformation at pH 10) and 300 nm (near the maximum positive difference of the extinction coefficients, $\triangle\varepsilon_{300} = 3.6$ mM$^{-1}$ cm$^{-1}$) for 4 h. The resulting absorption data were converted to conversions and equilibrium states as detailed in the Supplementary Information, giving the reaction courses and the relationship shown in Fig. 1j, k[21,39].

## Michaelis-Menten kinetics with *Gt*PNP

To evaluate the affinity of *Gt*PNP toward **X** under alkaline conditions (which essentially exclusively give **N7X**), we obtained Michaelis-Menten kinetics of this transformation at pH 10. To this end, we performed glycosylation reactions with **X** (20 − 500 µM), 200 µM **Rib1P** and 100 µg mL$^{-1}$ *Gt*PNP (equivalent to ca. 3.8 µM) in 50 mM glycine/proline buffer (containing 50 mM of both buffer components) pH 10 in wells of a UV-transparent 96-well plate in a total volume of 200 µL at room temperature (ca. 21 °C). These reactions were started by addition of 100 µL reaction mixture (containing all reagents except **X**) to 100 µL appropriately diluted solution of **X**. All reactions were run in duplicate and repeated with duplicates run with 50 µg mL$^{-1}$ *Gt*PNP (1.9 µM), for a total of four replicates per concentration of **X**. Reaction progress was monitored at 300 nm ($\triangle\varepsilon_{300} = 3.6$ mM$^{-1}$ cm$^{-1}$) for 20 min. The absorption change over time was approximated by linear fitting and converted into observed rate constants and kinetic parameters as detailed in the Supplementary Information, giving the Michaelis-Menten plot shown in Fig. 1l.

## Synthesis of N7X/N9X by various nucleoside phosphorylases

To examine the ability of various NPs to synthesize **N7X**, we screened a small but diverse selection of NPs. To this end, we performed glycosylation reactions with 500 µM **X**, 500 µM **Rib1P**, and 1 g L$^{-1}$ NP in 50 mM glycine-KOH buffer at pH 9 or 50 mM HEPES buffer at pH 7. These reactions were started by the addition of the enzyme and incubated at 50 °C for 1 or 24 h. Afterwards, a 50 µL sample of the reaction mixture was quenched by addition to 50 µL ice cold methanol and analyzed by HPLC as described above. For this experiment, we used bacterial (purine and pyrimidine NP from *Geobacillus thermoglucosidasius*, purine NP from *Escherichia coli*, and purine NP1 and NP2 from *Thermus thermophilus*) and mammalian NPs (human purine NP). These reactions yielded the conversions listed in Fig. 1m and Supplementary Table 5.

**Reporting summary**

Further information on research design is available in the Nature Portfolio Reporting Summary linked to this article.

## Data availability

All raw and calculated data underlying this study are freely available from the externally hosted supplementary information at zenodo.org (https://doi.org/10.5281/zenodo.8382795)[38]. Specifically, this includes UV, NMR, HRMS, and DFT results.

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

## Acknowledgements

The authors thank Samantha Voges (TU Berlin, NMR Department) for assistance with spectroscopic experiments and Dr. Maria Schlangen and Marc Griffel (TU Berlin, Center for Mass Spectrometry) for HRMS analyses. F.K. gratefully acknowledges funding by the Deutsche Forschungsgemeinschaft (DFG, German Research Foundation), project number 492196858.

## Author contributions

Conceptualization, A.K. and F.K.; Data curation, S.W., F.B., and F.K.; Formal analysis, S.W., F.B., S.K., and F.K.; Funding acquisition, A.K. and F.K.; Investigation, S.W., F.B., L.N., C.B., A.D., S.K., and F.K.; Methodology, F.K.; Project administration, A.K., F.K.; Resources, S.K., C.R.J., P.N., A.K., and F.K.; Software, -; Supervision, P.N., C.R.J., A.K., F.K.; Validation, -; Visualization, F.K.; Writing—original draft, F.K.; Writing—review & editing, all authors. Specifically regarding the experiments, A.D. and F.K. made the initial discovery. S.W. cloned the deaminase and the previously not reported nucleoside phosphorylases. S.W., L.N., C.B., A.D., and F.K. produced and purified the enzymes. S.W., L.N., C.B., and F.K. performed and analyzed the HPLC experiments (including the kinetics for the pH-dependent formation of **N7X**) and purified **N7X**. F.K. performed and analyzed the UV-spectroscopic experiments and carried out the physicochemical analyses. S.K. and F.K. carried out, and evaluated the NMR spectroscopic experiments. F.B. performed and evaluated the DFT calculations.

## Funding

## Competing interests

The authors declare no competing interests.
