## [Peer Review File · Nature Communications]

Nucleoside Phosphorylases Make N7-XanthosineREVIEWER COMMENTS

Reviewer #1 (Remarks to the Author):

The research described here by Westarp et al elegantly showcases wild-type nucleoside phosphorylases that indeed catalyze the regioselective coupling of xanthine and ribose 1-phosphate to N7-xanthosine instead of to N9-xanthosine, which would have previously been the expected product. The researchers then offer strong supporting evidence to corroborate the unexpected formation of N7-xanthosine as well as some atypical physicochemical properties of this molecule. I believe this work is of sufficient novelty and general interest to the greater scientific community to be worthy of publication in the prestigious Nature Communications, however with some proposed minor revisions.

1. As this is a preliminary communication, I submit that the scope of this work does not include the isolation and full characterization of N7-xanthosine; however, I suggest semi-preparative HPLC of the reaction mixture be re-run with solvents that contain either 0.1% TFA or 0.1% formic acid. I believe that doing so would narrow the elution time of the product and very likely destroy Rib1P but not N7X, offering a purer analyte. Further, was there a reason MeOH was used over acetonitrile? MeCN was utilized during analytical HPLC; why the change? Perhaps, using MeCN would delay the elution of N7X just long enough to allow the two major impurities to wash out first. Addressing these issues would provide strength to your method as a means to access this interesting nucleoside at larger scales, and thus offer more interest to the scientific community. Furthermore, an actual UV spectrum could be obtained with facility.

2. In line with point 1, more NMR data should be included. The authors intimate that a pure sample for NMR could not be obtained as a result of contamination with Rib1P and HEPES; however, perhaps with some tweaking of the gradient this could be resolved. Even if not, (the presentation of) a couple more experiments is warranted. In the main text, partial or complete ^1H and ^{13}C NMR spectra should be included with the spectra of the N9-regioisomer overlaid so the readers can better make the comparison between the two. This could offer preliminary insight into general trends in the NMR data between N7 and N9 nucleosides to hasten the assignment of other “non-native” nucleosides yet to be discovered. Such trends could further help underpin the reasons for the formation of the N7 regioisomer in the first place (electron density mapping, etc), a logical next step in the

research. In the supplementary information, Tables S1 and S2 should be expanded to include the N9 data as well to further facilitate a side-by-side comparison of the two regioisomers. Finally, it is customary to include all analyzed spectra that support your findings in the SI, not just as the crude acquired files (although certainly appreciated). Should these two points be addressed, I welcome this manuscript for publication in Nature Communications.

Finally, I commend the authors on the readability, conciseness and quality of the manuscript writing, both generally and in the English language. Nonetheless, there were some definite articles added in a couple of lines before a noun that would not usually have one:

- Page 1, Line 16 "... from the primary metabolism" should be changed to "from primary metabolism." This appears later in the manuscript too.
- Another minor point, Page 1, Line 37, I believe "a 'non-native' glycosylation regioisomer" is more appropriate than "the 'non-native' glycosylation regioisomer" as in principle, O-glycosylation and glycosylation at N1 and N3 could still take place (and has certainly been described chemically on other nucleobases).

Reviewer #2 (Remarks to the Author):

PNPs are key enzymes of nucleic acid metabolism. Exploring the capabilities and limitations of these enzymes is indeed a significant task for both modern enzymology and the synthetic chemistry. The article authors are well known for diligently adding to the pool of knowledge in this field for over a decade. Present study is on the same high level of quality as their other works.

The substrate properties of xanthine and xanthosine with respect to NPs have been periodically examined by various groups of researchers, but never in such detail. Indeed, the formation of purine nucleosides with an N-glycoside bond at the N7 position of the heterocyclic base in the presence of NPs has never, at least to my knowledge, been reported in the scientific literature before.

The methodology of the study is clear and straightforward, and generally meets the standards of research in the field. Below are some of the points addressing which would improve the overall quality of the work:

- Based on my prior work and experience with these specific compounds and enzymes, I was able to confirm the authors' conclusions on the structure of the N7X stereoisomer of xanthosine upon reviewing authors' spectra. However, in my opinion, the lack of complete isolation of N7X, or of obtaining its pure standard by chemical synthesis, is a drawback of the work. The wording used by the authors themselves in the discussion of the proof of the structure of the obtained compound, i.e. "partially purified", fully corresponds to reality. All the spectra presented contain signals associated with impurities, which complicate considerably the identification of the structure.

- The peculiarities of physical and chemical properties of N7X presented by the authors are rather interesting, but in my opinion, such interesting and important features should've been more reliably validated using pure, isolated, samples. For example, I would've done my best to verifiably exclude the presence of uric acid in the analyzed samples as, unfortunately, quite often the recombinant NP preparations contain an impurity of xanthine oxidase, and, thus, uric acid (which could also lead to bathochromic shift in UV spectra) is detected in reaction mixtures.

The availability of a pure standard would further facilitate the acquisition of thermodynamic and kinetic data and make it even more accurate.

- Considering the authors' global conclusions about the imperfection and even, and I quote, "promiscuity" of NPs, I cannot help but stand up for the "honor" of the enzymes.

The authors emphasize themselves, that the studies were conducted under non-standard and far from physiological conditions (pH, concentrations of reagents). Thus, although the results obtained by the authors undoubtedly contribute very valuable information to the general picture of NPs functioning and their interaction with substrates, they hardly indicate a critical violation of their functions as fine and precise catalysts, but rather can be regarded as interesting and unexpected features that are not critical in the metabolic cycle.

We wish to thank both reviewers for their time, effort and sharp eyes in reviewing our manuscript. We believe that the manuscript has benefitted greatly from their input. We have adopted all of their suggestions and expanded the SI accordingly. The additional data are now available from the corresponding zenodo entry.

During the revision, it was also brought to our attention that similar (although much less pronounced) redshifts between purine glycosylation isomers have been reported for analogues of 8-azaguanosine. Thus, we added a reference to this work in the revised manuscript (now ref. 21) and additionally referenced a related precedent in the SI (supplementary ref. 22).

Reviewer #1:

The research described here by Westarp et al elegantly showcases wild-type nucleoside phosphorylases that indeed catalyze the regioselective coupling of xanthine and ribose 1-phosphate to N7-xanthosine instead of to N9-xanthosine, which would have previously been the expected product. The researchers then offer strong supporting evidence to corroborate the unexpected formation of N7-xanthosine as well as some atypical physicochemical properties of this molecule. I believe this work is of sufficient novelty and general interest to the greater scientific community to be worthy of publication in the prestigious Nature Communications, however with some proposed minor revisions.

Thank you for this positive assessment.

1. As this is a preliminary communication, I submit that the scope of this work does not include the isolation and full characterization of N7-xanthosine; however, I suggest semi-preparative HPLC of the reaction mixture be re-run with solvents that contain either 0.1% TFA or 0.1% formic acid. I believe that doing so would narrow the elution time of the product and very likely destroy Rib1P but not N7X, offering a purer analyte. Further, was there a reason MeOH was used over acetonitrile? MeCN was utilized during analytical HPLC; why the change? Perhaps, using MeCN would delay the elution of N7X just long enough to allow the two major impurities to wash out first. Addressing these issues would provide strength to your method as a means to access this interesting nucleoside at larger scales, and thus offer more interest to the scientific community. Furthermore, an actual UV spectrum could be obtained with facility.

Indeed, N7-xanthosine proved difficult to purify, primarily due to its limited retention on reverse-phase materials and extreme streaking on silica. Following your suggestion, we re-purified our sample of N7-xanthosine and provide ¹H NMR data of this sample in the revised SI (Figure S6). These data provide useful insights into the multiplicity of H-5' and confirm the nature of H-1 and H-3 as exchangeable protons; Tables S2 and S3 have been updated and expanded accordingly. We did not obtain additional UV data on this material as the same problem of (in-)accurate weighing of small quantities still applies. We believe that our original approach of using isosbestic normalization based on a standard with accurately determined concentration is the best way to obtain the extinction coefficients of N7-xanthosine, as described in the SI.

2. In line with point 1, more NMR data should be included. The authors intimate that a pure sample for NMR could not be obtained as a result of contamination with Rib1P and HEPES; however, perhaps with some tweaking of the gradient this could be resolved. Even if not, (the presentation of) a couple more experiments is warranted. In the main text, partial or complete ¹H and ¹³C NMR spectra should be included with the spectra of the N9-regioisomer overlaid so the readers

can better make the comparison between the two. This could offer preliminary insight into general trends in the NMR data between N7 and N9 nucleosides to hasten the assignment of other “non-native” nucleosides yet to be discovered. Such trends could further help underpin the reasons for the formation of the N7 regioisomer in the first place (electron density mapping, etc), a logical next step in the research. In the supplementary information, Tables S1 and S2 should be expanded to include the N9 data as well to further facilitate a side-by-side comparison of the two regioisomers. Finally, it is customary to include all analyzed spectra that support your findings in the SI, not just as the crude acquired files (although certainly appreciated).

Following your suggestion, we recorded additional reference data for xanthine (**X**) and xanthosine (**N9X**) and their sodium salts in D₂O. We added a supplementary table comparing key ¹H and ¹³C chemical shifts of the nucleobase moieties of **X**, **N9X** and **N7X** (Table 4) and added a brief discussion and reference to this table to the main text. Indeed, compared to **N9X**, **N7X** does display notably different characteristic shifts of its anomeric proton and H-8 on the nucleobase, as well as the carbon centers comprising the imidazole unit. We agree that these trends deserve being highlighted and wish to thank you for prompting us to obtain additional data to this end.

Should these two points be addressed, I welcome this manuscript for publication in Nature Communications.

Finally, I commend the authors on the readability, conciseness and quality of the manuscript writing, both generally and in the English language. Nonetheless, there were some definite articles added in a couple of lines before a noun that would not usually have one:

- Page 1, Line 16 “... from the primary metabolism” should be changed to “from primary metabolism.” This appears later in the manuscript too.
- Another minor point, Page 1, Line 37, I believe “a ‘non-native’ glycosylation regioisomer” is more appropriate than “the ‘non-native’ glycosylation regioisomer” as in principle, O-glycosylation and glycosylation at N1 and N3 could still take place (and has certainly been described chemically on other nucleobases).

Thank you, we appreciate your kind words and highly constructive comments. We have implemented your suggestions for typographical improvements.

Reviewer #2:

PNPs are key enzymes of nucleic acid metabolism. Exploring the capabilities and limitations of these enzymes is indeed a significant task for both modern enzymology and the synthetic chemistry. The article authors are well known for diligently adding to the pool of knowledge in this field for over a decade. Present study is on the same high level of quality as their other works. The substrate properties of xanthine and xanthosine with respect to NPs have been periodically examined by various groups of researchers, but never in such detail. Indeed, the formation of purine nucleosides with an N-glycoside bond at the N7 position of the heterocyclic base in the presence of NPs has never, at least to my knowledge, been reported in the scientific literature before. The methodology of the study is clear and straightforward, and generally meets the standards of research in the field. Below are some of the points addressing which would improve the overall quality of the work:

Thank you for this positive assessment.

- Based on my prior work and experience with these specific compounds and enzymes, I was able to confirm the authors' conclusions on the structure of the N7X stereoisomer of xanthosine upon reviewing authors' spectra. However, in my opinion, the lack of complete isolation of N7X, or of obtaining its pure standard by chemical synthesis, is a drawback of the work. The wording used by the authors themselves in the discussion of the proof of the structure of the obtained compound, i.e. "partially purified", fully corresponds to reality. All the spectra presented contain signals associated with impurities, which complicate considerably the identification of the structure.

Following your suggestion and the constructive comments by reviewer 1, we re-purified our sample of N7-xanthosine and provide ¹H NMR data of this sample in the revised SI (Figure S6). These data provide useful insights into the multiplicity of H-5' and confirm the nature of H-1 and H-3 as exchangeable protons; Tables S2 and S3 have been updated and expanded accordingly. We also obtained additional comparative data for X and N9X in D₂O and added a comment on key spectroscopic characteristics of N7X in the main text.

- The peculiarities of physical and chemical properties of N7X presented by the authors are rather interesting, but in my opinion, such interesting and important features should've been more reliably validated using pure, isolated, samples. For example, I would've done my best to verifiably exclude the presence of uric acid in the analyzed samples as, unfortunately, quite often the recombinant NP preparations contain an impurity of xanthine oxidase, and, thus, uric acid (which could also lead to bathochromic shift in UV spectra) is detected in reaction mixtures. The availability of a pure standard would further facilitate the acquisition of thermodynamic and kinetic data and make it even more accurate.

Thank you for highlighting this potential issue to us. Indeed, xanthine oxidase impurities (oxidizing xanthine to uric acid) could have caused similar redshifts in the absorption spectra as the X→N7X transformation, which would have been a considerable source of error in our kinetic and thermodynamic data. Although the presence of xanthine oxidase in our preparations of heat-treated nucleoside phosphorylase is unlikely, we performed additional control experiments. Incubating inosine or xanthine with GfPNP under representative assay conditions used for the experiments in the manuscript gave no discernible redshift over the course of one hour (Fig. S5), indicating that no oxidation product was formed. We therefore conclude that any traces of potential xanthine oxidase activity do not possess meaningful levels

of activity. The equilibration of the glycosylation reactions monitored by UV (Fig. 1j) and HPLC (Fig. 1m) confirm this conclusion.

The purity of our sample of **N7X** did not impact the accuracy of any of the measurements of physicochemical parameters. The glycosylation reactions (Figs. 1j–l) were performed starting from (commercial, pure) **X** and the titrations for the determination of the pKa values are inherently independent of concentration and analyte purity by only considering the observed, UV-active species **N7X/N7X⁻**.

- Considering the authors' global conclusions about the imperfection and even, and I quote, "promiscuity" of NPs, I cannot help but stand up for the "honor" of the enzymes. The authors emphasize themselves, that the studies were conducted under non-standard and far from physiological conditions (pH, concentrations of reagents). Thus, although the results obtained by the authors undoubtedly contribute very valuable information to the general picture of NPs functioning and their interaction with substrates, they hardly indicate a critical violation of their functions as fine and precise catalysts, but rather can be regarded as interesting and unexpected features that are not critical in the metabolic cycle.

We appreciate your concern for the "honor" of these enzymes and agree – perhaps it would be appropriate to clarify. Following your suggesting, we have added some additional discussion and clarification in the closing remarks.

REVIEWERS' COMMENTS

Reviewer #1 (Remarks to the Author):

The authors have fully addressed my suggested revisions. I believe this manuscript now suitable for publication in Nature Communications.

Reviewer #2 (Remarks to the Author):

Thanks to the high standards of the authors in conducting original research and their responsive attitude to the gentle criticism and advice of the reviewers, the revised version of the paper looks complete and whole. Careful approach to correction of the paper's narrow points highlighted by reviewers allowed to refine the quality of the paper.

I would like to emphasize again the special and extensive contribution of the authors to the store of knowledge on key enzymes of nucleic acid metabolism, the high quality of their research and wish them new achievements.

Dr. Irina Varizhuk

Engelhardt Institute of Molecular Biology

Russian Academy of Sciences

We wish to thank both reviewers for their time and effort in reviewing our revised manuscript.

Reviewer #1:

The authors have fully addressed my suggested revisions. I believe this manuscript now suitable for publication in Nature Communications.

Thank you for this positive assessment.

Reviewer #2:

Thanks to the high standards of the authors in conducting original research and their responsive attitude to the gentle criticism and advice of the reviewers, the revised version of the paper looks complete and whole. Careful approach to correction of the paper's narrow points highlighted by reviewers allowed to refine the quality of the paper. I would like to emphasize again the special and extensive contribution of the authors to the store of knowledge on key enzymes of nucleic acid metabolism, the high quality of their research and wish them new achievements.

Thank you for this positive assessment.